# Enhancement of Multimodal Microwave-Ultrasound Breast Imaging Using a Deep-Learning Technique

**DOI:** 10.3390/s19184050

**Published:** 2019-09-19

**Authors:** Vahab Khoshdel, Ahmed Ashraf, Joe LoVetri

**Affiliations:** Department of Electrical and Computer Engineering, University of Manitoba, Winnipeg, MB R3T 5V6 1, Canada; Ahmed.Ashraf@umanitoba.ca (A.A.); joe.lovetri@umanitoba.ca (J.L.)

**Keywords:** microwave breast imaging, inverse problems, image reconstruction, tumor detection, convolutional neural networks

## Abstract

We present a deep learning method used in conjunction with dual-modal microwave-ultrasound imaging to produce tomographic reconstructions of the complex-valued permittivity of numerical breast phantoms. We also assess tumor segmentation performance using the reconstructed permittivity as a feature. The contrast source inversion (CSI) technique is used to create the complex-permittivity images of the breast with ultrasound-derived tissue regions utilized as prior information. However, imaging artifacts make the detection of tumors difficult. To overcome this issue we train a convolutional neural network (CNN) that takes in, as input, the dual-modal CSI reconstruction and attempts to produce the true image of the complex tissue permittivity. The neural network consists of successive convolutional and downsampling layers, followed by successive deconvolutional and upsampling layers based on the U-Net architecture. To train the neural network, the input-output pairs consist of CSI’s dual-modal reconstructions, along with the true numerical phantom images from which the microwave scattered field was synthetically generated. The reconstructed permittivity images produced by the CNN show that the network is not only able to remove the artifacts that are typical of CSI reconstructions, but can also improve the detectability of tumors. The performance of the CNN is assessed using a four-fold cross-validation on our dataset that shows improvement over CSI both in terms of reconstruction error and tumor segmentation performance.

## 1. Introduction

In quantitative microwave imaging, scattered-field data arising from the illumination of an object-of-interest (OI) by microwave energy is acquired via a data-acquisition setup. From the collected scattered-field data one then seeks to infer, or *reconstruct*, information about the spatial distribution of the physical properties of the OI that affect the scattering process, i.e., its permittivity. The reconstruction is in the form of two quantitative images corresponding to the real and imaginary parts of the complex-valued permittivity. Mathematically, the so-called inversion of the scattered-field data to reconstruct an image of the physical properties is accomplished by solving an *inverse problem*. Typically, a numerical forward model that relates any particular distribution of properties to the collected data is introduced into the inverse problem. The more general the reconstruction model the more difficult solving the inverse problem becomes, often leading to what are known as ill-posed inverse problems. Ill-posedness means that there may be no or many possible distributions of the properties that lead to the data (i.e., non-existence and non-uniquenes of solutions to the mathematical inverse problem), and that small changes in the data may lead to arbitrarily large changes in the inferred distributions of the properties (i.e., instability of the solution). Note that ill-posedness is a property of the mathematical formulation of the inverse problem corresponding to a particular setup of the wavefield process. One may attempt to change the physical process being used to collect the scattered-field data but ultimately the ill-posedness arises because of the limited amount of data one is able to collect as well as the fact that the data must be collected remotely, i.e., away from the location of the properties one is wanting to infer.

This is indeed the case for the inverse problems associated with quantitative microwave imaging (MWI) [1] and ultrasound imaging (USI) [2,3]. Both MWI and USI attempt a quantitative reconstruction of the spatial distribution of the physical properties affecting the wavefield processes associated with each of these modalities. In MWI one seeks to reconstruct the complex-valued permittivity of the OI, whereas in USI one reconstructs, e.g., the complex-valued compressibility and the mass-density of the OI. It has long been recognized that the reconstructions of physical properties, such as permittivity, would be extremely useful for medical imaging since different kinds of biological tissues have been shown to have distinct electrical properties. For the detection of breast cancer significant effort has gone into the characterization of breast tissues [4].

There have been significant advances in the use of microwave imaging for breast cancer detection and monitoring during the past two decades [5,6]. Microwave breast imaging techniques generally split into those that produce a qualitative image of the breast’s interior structure and those that produce a quantitative image of the complex permittivity of the breast tissues. Quantitative techniques take advantage of the different dielectric properties of normal breast tissue (e.g., skin, adipose, and fibroglandular) and cancerous tumors [4,7]. The main approach for solving ill-posed inverse problems associated with quantitative MW/US breast imaging are computationally expensive iterative methods where the inversion model consists of a numerical solution of a complex electromagnetic or ultrasound scattering problem [3,6]. While such iterative methods have improved dramatically over the years, providing improved resolution and accuracy of the reconstructed properties, as well as more efficient implementations, there are still many fundamental trade-offs between these three aspects due to operational, financial, and physical constraints [6]. The greatest challenge to MWI becoming clinically accepted for breast imaging is its lower resolution compared to other biomedical modalities as well as the many reconstruction artifacts that are produced related to the nonlinearity and ill-posedness of the associated inverse problem. Progress in MWI has recently been made by incorporating prior information obtained from some alternate modality. For instance, it has recently been shown that spatial segmentation priors derived from magnetic resonance imaging or Xray computed tomography can significantly improve MWI [8]. In this paper we take advantage of the recent progress in a multimodal ultrasound/microwave imaging procedure wherein the breast region is first segmented into tissue regions using an ultrasound (US) technique and subsequently these tissue-regions are incorporated as prior information into either a qualitative, [9], or quantitative microwave imaging technique [10].

Deep learning is currently an extremely active research area that has produced huge successes in a broad range of applications such as speech recognition, computer vision, and natural language processing [11]. Researchers have begun to investigate how deep learning techniques can be effectively applied in biomedical imaging and inverse problem [12,13]. Studies have shown that deep learning can provide state-of-the-art performance for tumor classification [14], segmentation [15], and the post-processing of X-ray CT images [16,17].

Neural networks have recently been combined with microwave image construction techniques as a means of learning the forward model for a complex data-acqusition system [18]. Also, Rekanos has proposed using radial basis function neural networks to solve a simple inverse problem associated with the microwave imaging [19]. In this technique they estimate the position and size of proliferated marrow inside bone. Most recently, Li et al. have studied how deep neural networks can be used to take microwave images created using the back-projection (BP) method and have the network output a much improved image [20]. This by-passes the use of iterative techniques to solve the full nonlinear electromagnetic inverse problem as back-projection is a linear construction technique. The results of that work are significant because they show how a network trained using only BP images obtained from synthetically generated data can then generalize to BP images created using experimental data. Although the generalization to experimental data is significant, the targets they used are simple homogeneous targets with low contrast. That is, the targets used by them for training the network had no internal structure. To the best of our knowledge utilizing deep learning techniques to enhance microwave imaging for high contrast objects-of-interest that have complicated internal structure, such as the breast imaging application that we consider herein, have not been previously studied.

In this paper, we explore one method of utilizing deep learning in dual-mode MW/US breast imaging that utilizes the Contrast Source Inversion (CSI) technique. In addition, we quantify the developed framework’s ability to categorize and/or identify tissues in the images. The specific goal is to identify tumors within breast images of the reconstructed real and imaginary parts of the complex valued permittivity.

## 2. CSI-Deep-Learning Microwave Breast Imaging

### 2.1. Contrast Source Inversion

The first part of the CSI-Deep-Learning technique developed herein is to perform preliminary (rough) reconstructions of the complex-valued permittivity using the CSI technique. Traditionally, regularized iterative algorithms have remained the method of choice for solving inverse problems associated with MW breast imaging. Since its creation in 1997 the CSI technique has become a leading numerical optimization based reconstruction method for inverse problems [21]. The method can be briefly described as follows for the type of 2D Transverse-Magnetic tomographic reconstructions considered herein. We define an imaging domain, D, within the problem domain Ω of a MW or US (wavefield) imaging setup as shown in Figure 1. An unknown isotropic, nonmagnetic OI is located in D and is surrounded by a background medium of known electrical and acoustic properties. The complex relative permittivity of the OI is $\epsilon_{r}\left(r\right)$, where r is a 2D position vector. The corresponding electric contrast is defined as $\chi \left(r\right)≜\left(\epsilon_{r}\left(r\right)-\epsilon_{n}\left(r\right)\right)/\epsilon_{n}\left(r\right)$ where $\epsilon_{n}\left(r\right)$ is an imposed numerical background complex relative permittivity (with χ(r)=0 for r∉D), which may be inhomogeneous so as to represent any prior information we wish to impose (notice that at any location where $\epsilon_{n}\left(r\right)=\;\epsilon_{r}\left(r\right)$ the contrast will be zero). With the OI in D a total field $E_{t}$ is produced by a source *t* and is measured at points located on a measurement surface S. An incident field $E_{t}^{\mathrm{inc}}$ is defined when there is no OI in D; rather the region is occupied by the imposed (known) numerical background $\epsilon_{n}\left(r\right)$. The scattered electric field is then defined by $E_{t}^{\mathrm{sct}}≜E_{t}-E_{t}^{\mathrm{inc}}$ and satisfies the scalar Helmholtz equation $\nabla^{2}E_{t}^{\mathrm{sct}}\left(r\right)+k_{n}^{2}\left(r\right)E_{t}^{\mathrm{sct}}\left(r\right)=-k_{n}^{2}\left(r\right)w_{t}\left(r\right)$ where $k_{n}\left(r\right)=\omega \sqrt{\mu_{0}\epsilon_{0}\epsilon_{n}\left(r\right)}$ is the wavenumber in the imposed numerical background, and $w_{t}\left(r\right)≜\chi \left(r\right)E_{t}\left(r\right)$ is the contrast source that produces the scattered field for that transmitter.

The boundary-value problem (BVP) defined by this second-order Partial Differential Equation (PDE) and the boundary conditions is solved using the Finite-Element Method (FEM) [22]. Thus, the problem domain (Ω) is divided into a mesh of triangular elements upon which linear-basis functions are specified whose parameters are dependent only on the geometry of the mesh. Applying FEM to the BVP discretizes the problem producing a matrix equation for the unknown scattered-field in terms of the contrast source (for each transmitter *t*).

The inverse problem is that of minimizing the CSI cost-functional that is defined with respect to the contrast sources, $w_{t}$, and the contrast, χ. This functional is constructed as the sum of normalized data-error and domain-error functionals written as
(1)$$\begin{array}{c} \begin{array}{cc} F^{\mathrm{CSI}}\left(\chi ,w_{t}\right) & =F^{S}\left(w_{t}\right)+F^{D}\left(\chi ,w_{t}\right)\\ & =\frac{\sum_{t}{∥f_{t}-M_{S}L\left[w_{t}\right]∥}_{S}^{2}}{\sum_{t}{∥f_{t}∥}_{S}^{2}}\;+\\ & \frac{\sum_{t}{∥\chi ⊙E_{t}^{\mathrm{inc}}-w_{t}+\chi ⊙M_{D}L\left[w_{t}\right]∥}_{D}^{2}}{\sum_{t}{∥\chi ⊙E_{t}^{\mathrm{inc}}∥}_{D}^{2}} \end{array} \end{array}$$

For each transmitter *t*, $f_{t}$ holds the measured scattered-field data at the receiver locations, $E_{t}^{\mathrm{inc}}$ is the vector of incident field mesh values inside D. The matrix L is the inverse of the FEM matrix operator that transforms contrast source variables $w_{t}$ in D to scattered field values within the whole domain Ω. The operator $M_{S}$ transforms field values from Ω to receiver locations on the measurement surface S whereas the operator $M_{D}$ transforms values from Ω to points inside D. The vector χ holds the contrast over the FEM mesh nodal values located inside D. The notation a⊙b denotes the Hadamard (i.e., element-wise) product. The CSI objective functional $F^{\mathrm{CSI}}\left(\chi ,w_{t}\right)$ is minimized by updating the contrast source $w_{t}$ and the contrast χ variables sequentially.

On its own, CSI is not able to effectively reconstruct the high-contrast complex-valued permittivities associated with breast tissues, but recent advances have shown that incorporating prior information regarding the location of specific tissue regions into CSI as an inhomogeneous numerical background leads to much improvement [23,24]. Even so, the algorithm still produces imaging artifacts that make further image analysis difficult. In [25], it was shown how US-derived prior tissue regions can greatly improve reconstructions based on qualitative radar techniques. Herein, we use these US-derived tissue regions as prior information for CSI-based MWI.

### 2.2. Machine Learning Approach to Reconstruction

One of the main reasons for imaging artifacts in CSI reconstructions is the possibility of a non-realistic map of permittivities producing realistic measurement data. Any numerical optimization algorithm is thus prone to getting stuck in a bad local minima corresponding to non-realistic map of permittivities. As seen in the previous section, to discourage bad solutions, the CSI objective does incorporate a regularization term. However, even with regularization, artifacts in CSI outputs are common. In this paper we aim to fuse the CSI technique with a deep learning approach–our objective is to learn a data-driven mapping, G, from a rough CSI reconstruction to the true permittivity ($G:\epsilon^{CSI}\to \epsilon^{true}$). Since the permittivity values are complex, CSI output has a real and an imaginary part; so does the desired permittivity $\epsilon^{true}$. It is worth noting that the real and imaginary parts of the permittivity have very different scales, and a reconstruction loss to find the optimal G can easily get dominated by one of the components. Several normalization options can be considered; however, to simplify, we decompose the task of estimating G into two parts, $G_{R}$ and $G_{I}$. Essentially we learn two separate functions to reconstruct the real and imaginary parts of the permittivity. If the permittivity map is an M×N image, then each of the learned functions maps M×N complex domain to M×N real domain (e.g., $G_{R}:C^{M\times N}\mapsto R^{M\times N}$). The complex output of CSI can be treated as a 2-channel image. As such, to learn the two functions, the input and output are both images. In particular, we aim to realize $G_{R}$ and $G_{I}$ through deep neural networks.

### 2.3. Choice of Neural Network Architecture

Traditional convolutional neural networks (CNNs) were originally designed as classifiers–for input they would take in an image, and as output they would produce a decision or an object category. However, recent years have seen several variations in architectures for which the input and output are both images. The requirement for the neural network output to be an image often arises in applications such as image segmentation. For tumor segmentation, the U-Net architecture has shown very promising results [15]. The architecture consists of successive convolutional and downsampling layers, followed by successive deconvolutional and upsampling layers. Moreover, the skip connections between the corresponding contractive and expansive layers keep the gradients from vanishing that helps in the optimization process. To use the architecture for our reconstruction problem, we change the segmentation objective of the U-Net architecture, and replace it with the sum of pixelwise squared reconstruction errors between the true permittivity and the network output.

Note that our training data, which forms the input to the CNN, are complex valued images. Thus, a choice needs to be made on a CNN architecture that can accommodate complex values. Very little has been reported on the training of U-Net with complex weights, although recently there has been some work on training neural networks with complex weights for convolutional architectures [26]. To establish a proof-of-concept, in this paper we make the choice of testing two different U-Net architectures having real-valued weights, described as follows.

**Architecture 1:** The input to the first network consists of three CSI reconstructions after 250 iterations: we use the real and imaginary parts along with the magnitude of the complex-valued permittivity. The network is trained to output either the real or the imaginary part of the corresponding true permittivity map. That is, two independent networks, with three input images and one output image, are trained to reconstruct the real and imaginary parts of the permittivity.

**Architecture 2:** The input is same as the first architecture but the output consists of all three images corresponding to the magnitude, the real and imaginary parts of the true complex-valued permittivity. Hence, only one network with three inputs and three outputs is trained.

The schematic for the first architecture is shown in Figure 2.

### 2.4. Datasets

To create the synthetic scattered-field data for training and testing the CSI-deep-neural-network, Magnetic Resonance Imaging (MRI)-derived numerical breast phantoms are first created by converting each pixel value in tomographic slices of MRI breast images to a corresponding complex-valued permittivity. As base models, three different MRI-derived breast models are utilized as shown in Figure 3. Breast Model I, shown in the left column of Figure 3, is a heterogeneously dense breast of dimensions 10 cm × 9.5 cm with two tumors of approximate dimensions 1.9 × 2.3 cm and 1.9 × 1.8 cm centered at (0.9, 18.4) and (30.14, 2.8). Breast Model II, shown in the middle column, is a 11.2 cm × 9.6 cm fatty breast that contains one tumor of size 1.5 cm × 1.6 cm centered at (7.3, 3). Breast Model III is a very dense breast having dimensions of 10 cm × 5 cm and contains one tumor of size 2.3 cm × 1.6 cm centered at (4.1, 2.7). All tumors are within the fibroglandular tissue. Ultrasound scattered-field data is generated for each model and is utilized to generate prior information in the form of three tissue regions: a skin layer, a fat region, and the fibroglandular region (similar to what was done in [25]). This prior is utilized as an inhomogeneous numerical background in each of the CSI-based MWI reconstructions that constitute the datasets for training and testing the deep neural network. The forward data is obtained using a finite-element 2D electromagnetic field solver and 5% noise is added to the forward data before it is inverted using the FEM-CSI technique. Details can be found in [9].

In order to generate a sufficient number of examples having various tumors to train the neural networks considered herein, the permittivity values at the pixels where the tumors exist in the three breast models are first set to that of fibroglandular tissue. Then, for each breast model, 400 phantoms are generated with either a single or two tumors occupying the fibroglandular region. In particular 200 phantoms have one tumor and the other 200 are generated with two tumors. The generated tumors have an approximate maximum diameter of 1.1–1.5 cm and are randomly located within the fibroglandular region (contiguous pixels are randomly grown from a starting point until a maximum diameter is reached). Examples of numerical phantoms for each breast model are shown in the right column of Figure 4. A total of 400 phantoms were generated for each of the three models, so our dataset consists of 1200 breast phantoms with 600 examples each for single and two tumors. Microwave as well as ultrasound scattered-field data was generated for each of the 1200 phantoms and multimodal CSI reconstructions were performed.

## 3. Numerical Experiments

All the CNNs were implemented in Python 3.6 using the Keras library running Tensorflow as backend. We used a Tesla P100-PCIE-12GB graphic processor (NVIDIA Corporation, Santa Clara, CA, USA) and Intel (R) CPU (3.50 GHz) (Intel Corporation, Santa Clara, CA, USA). The convolutional layer weights were initialized by Gaussian random distribution using Xavier’s method to obtain an appropriate scale [27]. We used a batch size of 10 and ran training for 75 epochs.

### 3.1. U-Net Training and Quantitative Assessment

To assess the proposed methodology all experiments were performed utilizing a four-fold cross-validation strategy. Two types of training settings were performed differentiated by the breast models used for training. U-Net A was trained using images from all three types of breast models whereas U-Nets B, C, and D were trained using images from only two types of breast models. Details of the testing that was performed is as follows:
**Training Setting: U-Net A:** In this setting, training was done using examples from all three breast models. The 1200 images of the dataset were divided into 4 groups consisting of 300 images (100 from each breast model). To implement the four-fold cross-validation, four networks were trained, each using three groups for training (900 images) and tested using the remaining hold-out group (300 images). Thus all 1200 cases featured as test examples when they were not part of the training set.**Training Setting: U-Net B, U-Net C and U-Net D:** In this setting, three different U-Nets where breast examples from one type of breast model was excluded from the training set were trained. In U-Net B examples from breast Model III were excluded, in U-Net C examples from breast Model II were excluded, and in U-Net D examples from breast Model I were excluded from the training set. The training images were taken from the same groupings of the four-fold cross-validation used for U-Net A but now only 600 images from three of the groups were used for training. Testing was performed using images from the hold-out group. In addition testing was performed using all combinations of breast model from the hold-out group. That is, utilizing from either breast Model I, II, III, I & II, I & III, or II & III. The first three combinations consist of only 100 images while the latter three consist of 200 images from the hold-out group. This type of testing was motivated by the fact that each model is significantly different from the rest of the models; excluding them from the training set taxes the neural network when during testing it is presented with reconstructions from the unseen model.**Performance Metrics:** We quantitatively assess both the reconstruction capability and the tumor segmentation performance of the trained U-Nets. To assess the reconstruction quality, we use the Root Mean Squared (RMS) reconstruction error between the network output and the true permittivity values (for both the real and imaginary parts separately). To quantify the tumor segmentation performance, we use the Area Under the Curve (AUC) of the pixel-wise Receiver Operating Characteristics (ROC) using the reconstructed permittivity as a feature. Pixel-wise ROC-AUC is a good performance measure for tumor segmentation since it quantifies the separation between the distribution of permittivities of tumor and non-tumor pixels [28]. For comparison we computed RMS reconstruction error and performed ROC analysis on CSI-only reconstructions. The results of this quantitative evaluation are shown in Table 1 and Table 2.

### 3.2. Qualitative Evaluation of Robustness

To evaluate the robustness of the proposed methodology for creating the CNNs we check the sensitivity to the parameters of the training dataset. Specifically, the parameters that were used in generating the CSI images as well as the parameters used in presenting the images to the CNN. First, given that all CNNs were trained utilizing CSI images that were terminated after 250 iterations, the performance of the best performing trained network (U-Net A) is checked by testing with CSI reconstructions that were terminated after 20, 150 and 500 iterations. Comparative examples of output from U-Net A for these CSI images at these iteration numbers are shown in Figure 5 and Figure 6. This shows the performance of the CNN when tested against images that could vary substantially in terms of accuracy from the ones with which it was trained; a type of test against the proximity of the input to the test set.

An additional test regarding the required proximity of the input to the test set is to test the robustness of the network against geometric transformations of the input images such as rotation and flipping. Note that U-Net A was trained using only unrotated and unflipped images. Figure 7 demonstrates the result of the trained network when the input images were rotated and flipped.

## 4. Results and Discussion

Table 1 and Table 2 show the RMS reconstruction errors (cyan cells) and ROC-AUC (gray cells) for the different training and testing setups previously described. Figure 8 shows ROCs when the test sets included breast models I, II, III (plot (a) and (c)) and only Model II (plot (b) and (d)).

From these results we find that U-Net A, which was trained on examples from all breast models, performs better (lower reconstruction error and higher AUC) than the both CSI and other U-Nets: B,C,D. This is expected because the three breast models vary considerably in their internal structure. Although the sub-optimal networks in all cases improve the RMS error over that obtained using CSI, the AUC is sometimes improved but actually deteriorates for some of these networks.

It is expected that the CNN will perform worse when a whole class of breast models is not seen by the neural network during the training. From the results it can be observed that deterioration in the AUC is most pronounced in U-Net B (trained using Models I and II) and U-Net D (trained using Models II and III), when they are tested using the missing model. The same does not happen with U-Net C (trained using Models I and III). Its performance does not deteriorate substantially compared to U-Net A. It is too early to conclude from this observation that a U-Net trained on heterogeneously dense or very dense breast models can generalize to improve the CSI reconstructions of fatty breasts. It is possible that a thresholding technique might not be an effective test for detecting tumors which make up a large part of the fibroglandular tissue (as is the case for Model II breasts). This is because CSI on its own already detects the small amount of fibroglandular tissue in Model II breasts very well with few errors in the fatty tissue, thus, thresholding the image always detects a part of the fibroglandular tissue as tumor, and this has a good probability of being truly tumor (high true-positive rates). What this implies is that the thresholded ROC-AUC metric is not able to accurately report on the improvements U-Net is able to impart on the CSI reconstructions.

It is notable, however, that U-Net C is able to generalize and not corrupt Model II breasts (having never seen that type of breast). This is a surprising result and leads us to believe that a U-Net trained on a wider variety of breast models will be quite robust in providing improvements of CSI reconstructions.

In terms of robustness and generalization, although testing the networks on rotated and flipped images caused a drop in performance, the U-Net output remains much better than CSI. For qualitative visualization of results, representative examples of reconstruction are shown in Figure 7. This is also true when testing with CSI reconstructions that were stopped at different iterations, as can be seen in Figure 5 and Figure 6.

The other observation relates to the differences between Architectures 1 and 2. Using Architecture 2, there seems to be a slight improvement in the AUC over Architecture 1 when the ROC is obtained using the imaginary part of the complex permittivity. On the other hand the RMS error slightly deteriorates, which is not surprising because Architecture 2 only has both the real and imaginary parts as an output and therefore both affect the cost-function. Given that the magnitude of the real part is greater than that of the imaginary part, the real part dominates the cost-function. It is surprising, though, that the AUC improves when Architecture 2 is used. This leaves the differences between Architectures 1 and 2 inconclusive. More research is required to differentiate the differences between these two Architectures. In addition, future work will explore the potential use of architectures utilizing complex weights (see for example [26]).

## 5. Conclusions

A deep learning technique using a CNN that takes in tomographic reconstructions representing the complex-valued permittivity of the breast obtained using the CSI algorithm and produces images that are much closer to the true permittivity has been introduced. The CNN reconstructed images show the network’s ability to remove artifacts that are typical of CSI reconstructions. Not only is the RMS error between the CNN reconstructed images and the true permittivity images improved over the CSI reconstructions, but by using a simple thresholding detection algorithm tumors that are not detectable in CSI reconstructions are correctly detected in the CNN output. To assess the network’s generalization we have conducted experiments by testing on examples that differ from the training examples in varying degrees in terms of breast models and geometric transformations. Though our dataset consisted of numerically generated phantoms, we used numerical breast models derived from MRI tomographic images to generate the multimodal MW/US reconstructions. Three types of breast phantoms that span the array of BI-RADS Breast Densities were used for this study. It was found that training with all three breast-types produced the most robust CNN: consistently improving the AUC and RMS error metrics over the CSI results. It was found that when certain breast model-types were missing from the training set, sometimes the AUC actualy degrades whereas the RMS is still improved. Thus, a significant finding of this work is that a wide diversity of breast-types should be used in the training of CNNs for this purpose.

The first step in extending this work is to train 3D-CNNs that take full-3D CSI images as input. The excellent results also motivate future work that will involve the use of experimental scattered-field data derived from real breast multimodal MW/US imaging. Preliminary tumor-detection results using experimental phanotms in a unique experimental breast imaging system have recently been reported [29]. Future work will test the ability of CNNs trained with 3D images to improve the CSI results obtained from that system.

## Figures and Tables

**Figure 1 sensors-19-04050-f001:**
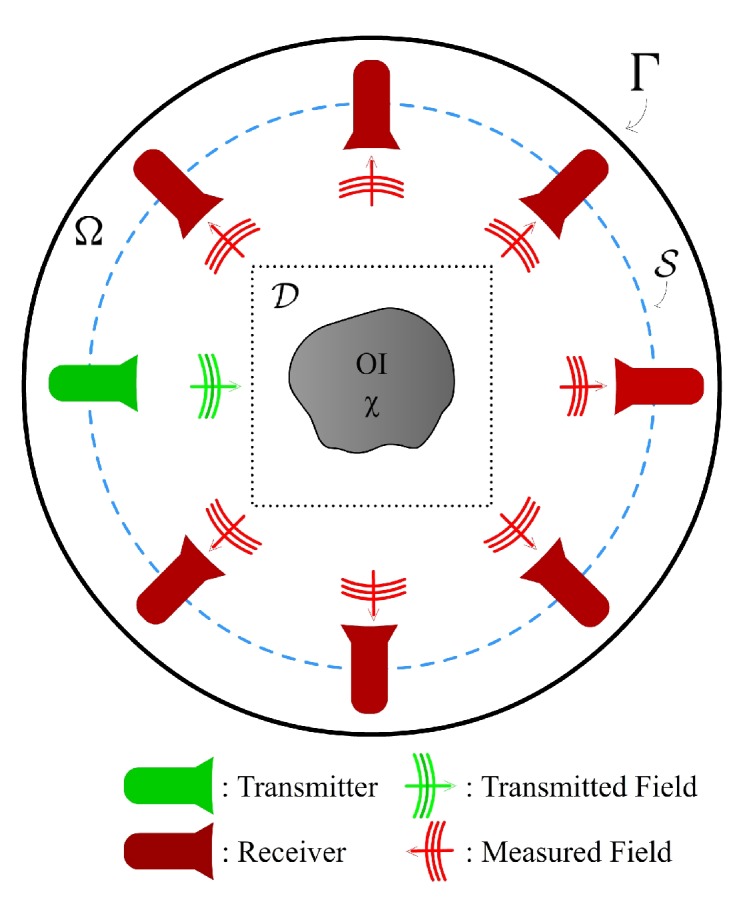
Schematic representation of a wavefield imaging setup.

**Figure 2 sensors-19-04050-f002:**
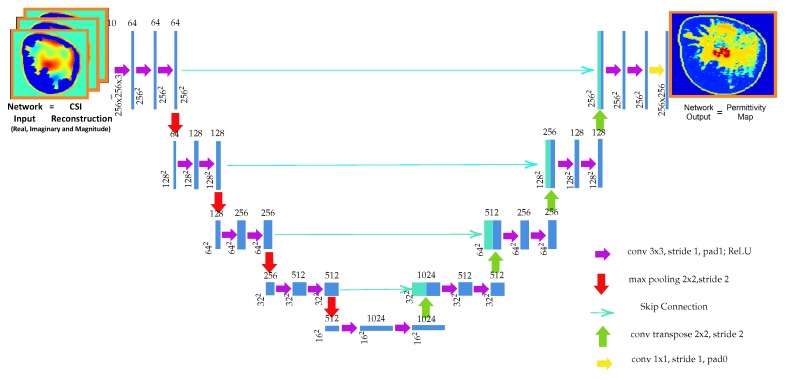
Schematic for the U-Net Architecture 1 for the proposed permittivity reconstruction. The input to the network is the CSI reconstruction, and the network is trained to output the corresponding true permittivity map. Two networks with the above architecture were trained to reconstruct the real and imaginary parts of the permittivity.

**Figure 3 sensors-19-04050-f003:**
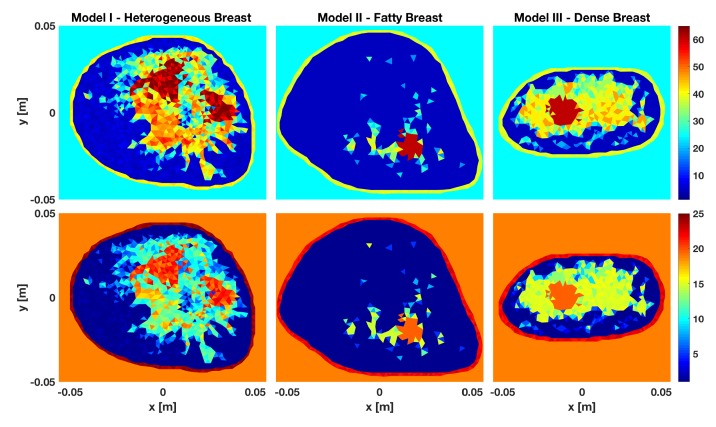
Three breast models: I - Heterogeneously dense breast, II - Fatty, III - Very dense breast. Top row-real part, bottom row-imaginary part of permittivity.

**Figure 4 sensors-19-04050-f004:**
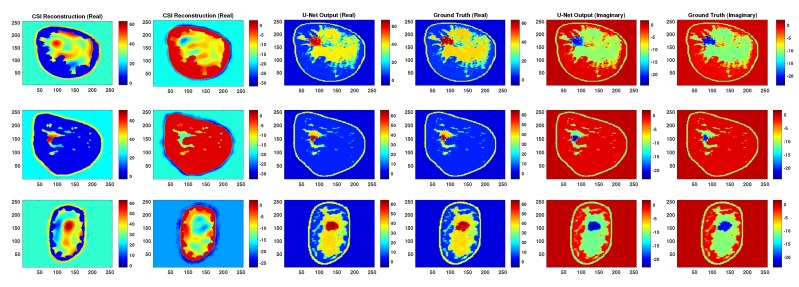
Representative reconstructions from each breast model for U-Net A setup (Architecture 1: Training set: Breast Models I, II, III; Testing set: Breast Models: I, II, III).

**Figure 5 sensors-19-04050-f005:**
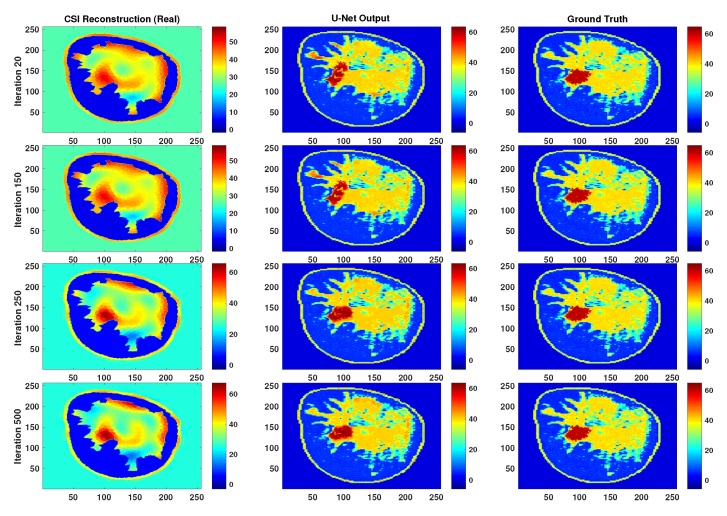
Representative reconstructions of the real part for a particular example when the test images were stopped at iteration number 20, 150 and 500 but the neural net was trained on iteration number 250.

**Figure 6 sensors-19-04050-f006:**
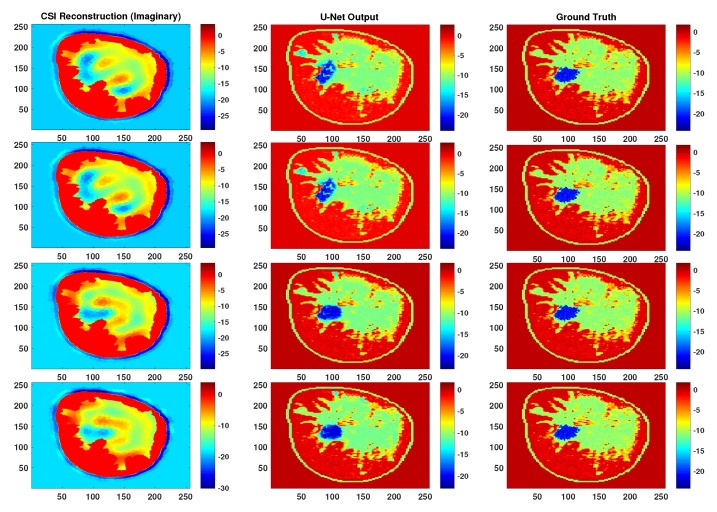
Representative reconstructions of the imaginary part for a particular example when the test images were stopped at iteration number 20, 150 and 500 but the neural net was trained on iteration number 250.

**Figure 7 sensors-19-04050-f007:**
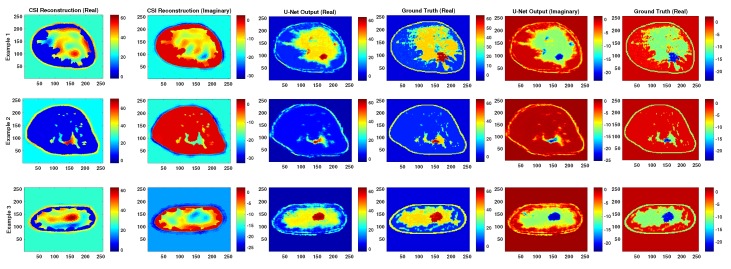
Representative reconstructions from each breast model when the test images were rotated and flipped but the neural net was trained on unrotated and unflipped images.

**Figure 8 sensors-19-04050-f008:**
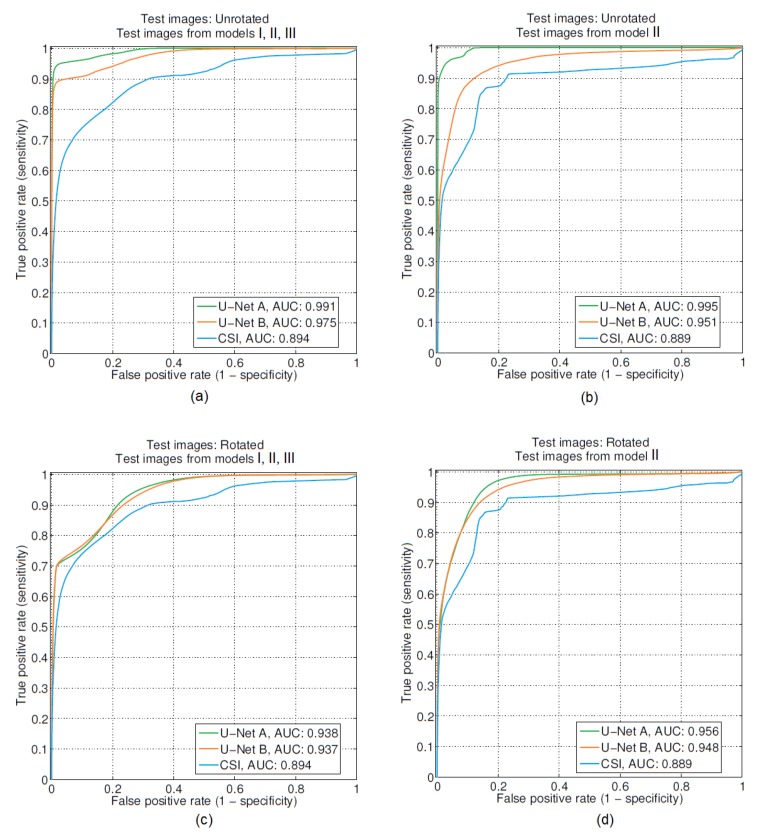
Comparison between the detection performance based on the reconstructed outputs of U-Net and CSI. For all the above cases U-Net A and U-Net B represent the settings when the network was trained on examples from models I, II, III and models I, III respectively. Training was always done on unrotated images. Testing scenarios (whether the images were rotated or not) as well as what models were included in testing are shown on top of each plot (**a**–**d**).

**Table 1 sensors-19-04050-t001:** Comparison of reconstruction and tumor detection performance for architecture 1. The entries in the cells show the reconstruction RMS error (Cyan) and ROC-AUC (Gray) for the respective setting. The top-half of the table shows the performance when the real part of the reconstructed permittivity was used to compute the reconstruction error and to generate the ROC curve and the bottom-half shows the performance related the imaginary part of the reconstructed permittivity.

Models Included in the Training Set	Reconstruction Technique	Breast Models Included in the Test Set						
		I, II, III	I	II	III	I, II	I, III	II, III
N/A	CSI	2.199	2.214	1.931	2.423	2.077	2.321	2.191
		0.897	0.868	0.892	0.897	0.890	0.882	0.915
I, II, III	U-Net A	0.122	0.144	0.075	0.135	0.114	0.140	0.110
		0.987	0.980	0.994	0.984	0.987	0.982	0.991
I, II	U-Net B	1.329	0.257	1.158	1.973	0.839	1.407	1.618
		0.596	0.908	0.653	0.442	0.871	0.593	0.387
I, III	U-Net C	0.694	0.253	1.151	0.232	0.834	0.243	0.830
		0.922	0.886	0.937	0.919	0.917	0.903	0.942
II, III	U-Net D	1.297	1.947	1.098	0.231	1.580	1.386	0.794
		0.758	0.587	0.730	0.929	0.631	0.740	0.904
N/A	CSI	7.103	6.894	6.548	7.806	6.723	7.364	7.205
		0.757	0.717	0.799	0.713	0.762	0.721	0.781
I, II, III	U-Net A	0.307	0.352	0.205	0.342	0.288	0.347	0.282
		0.987	0.981	0.992	0.985	0.987	0.983	0.990
I, II	U-Net B	1.884	0.594	1.572	2.797	1.188	2.022	2.268
		0.654	0.889	0.907	0.529	0.906	0.649	0.465
I, III	U-Net C	1.034	0.547	1.609	0.561	1.202	0.554	1.205
		0.913	0.888	0.937	0.895	0.912	0.894	0.929
II, III	U-Net D	1.843	2.754	1.512	0.567	2.221	1.989	1.142
		0.694	0.501	0.810	0.916	0.536	0.671	0.909

**Table 2 sensors-19-04050-t002:** Comparison of reconstruction and tumor detection performance for architecture 2. The results show that performance when the real and imaginary part of the reconstructed permittivity was used to compute the reconstruction error and to generate an ROC curve.

Models Included in the Training Set	Reconstruction Technique	Breast Models Included in the Test Set						
		I, II, III	I	II	III	I, II	I, III	II, III
N/A	CSI	2.199	2.214	1.931	2.423	2.077	2.321	2.191
		0.897	0.868	0.892	0.897	0.890	0.882	0.915
I, II, III	U-Net A	0.126	0.149	0.078	0.140	0.119	0.145	0.114
		0.988	0.982	0.993	0.985	0.988	0.983	0.991
I, II	U-Net B	1.310	0.268	1.087	1.973	0.792	1.408	1.593
		0.623	0.892	0.944	0.515	0.912	0.612	0.434
I, III	U-Net C	0.678	0.241	1.125	0.236	0.813	0.238	0.813
		0.921	0.894	0.949	0.917	0.917	0.905	0.938
II, III	U-Net D	1.285	1.941	1.061	0.242	1.565	1.383	0.770
		0.753	0.581	0.841	0.924	0.620	0.734	0.918
N/A	CSI	7.103	6.894	6.548	7.806	6.723	7.364	7.205
		0.757	0.717	0.799	0.713	0.762	0.721	0.781
I, II, III	U-Net A	0.313	0.361	0.206	0.349	0.294	0.355	0.286
		0.989	0.985	0.992	0.987	0.989	0.987	0.991
I, II	U-Net B	1.899	0.599	1.607	2.807	1.213	2.030	2.287
		0.670	0.896	0.938	0.516	0.913	0.643	0.516
I, III	U-Net C	1.044	0.561	1.630	0.546	1.219	0.553	1.215
		0.918	0.889	0.943	0.910	0.914	0.901	0.935
II, III	U-Net D	1.851	2.751	1.547	0.558	2.232	1.985	1.163
		0.728	0.566	0.819	0.951	0.600	0.703	0.920

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
