# Peer review of "Enhancement of Multimodal Microwave-Ultrasound Breast Imaging Using a Deep-Learning Technique"

_sensors, 2019, doi:10.3390/s19184050_

Round 1

Reviewer 1 Report

This paper, which is very well written, makes a step forward in the application of deep learning to inverse scattering and in particular to microwave tomography for breast cancer detection.

In particular, while somehow 'blind' NN (using raw data as input) or (more recently) backpropagation solutions are presently used as input to the network, this contribution suggests instead to exploit CSI based reconstructions. This is in my way the right way to proceed, as one is exploiting indeed all the huge amount of knowledge accumulated during the years in inverse scattering in order to reduce the complexity of the problem to be solved from the neural network.

The presented results are sound.

As minor comments :

i. there are a number of typos in the conclusions section. Please correct

ii. other 'multimodal' reconstruction techniques have been recently discussed in the literature based on initial segmentation by TAC or MRI followed by 'segmented' contrast source inversion, which strongly regularizes the problem. Some comments on these contributions by italian researchers could be of interest

 iii. A recent contribution by Xudong Chen [IEEE Trans on GRS, 2018] which is also using deep learning is also worth to be considered.

Author Response

We wish to thank the reviewer for taking the time to evaluate our paper and for the thoughtful and constructive comments that have helped us to improve the paper. We have tried to address your comments as follows.

Once again, many thanks for your kind considerations.

There are a number of typos in the conclusions section. Please correct,

Reply:We have now re-written the Conclusion in the hope that is more clear regarding what we have accomplished.

Other 'multimodal' reconstruction techniques have been recently discussed in the literature based on initial segmentation by TAC or MRI followed by 'segmented' contrast source inversion, which strongly regularizes the problem. Some comments on these contributions by Italian researchers could be of interest.

Reply: we have added a short new paragraph in the Introduction which refers to the MRI/XrayCT derived prior information work recently reported by the Italian group. In this same paragraph we also refer to two of our recent publications on the use of an Ultrasound/Microwave multimodal imaging technique.

A recent contribution by Xudong Chen [IEEE Trans on GRS, 2018] which is also using deep learning is also worth to be considered.

Reply:We have read this contribution and feel it is relevant so in the revised version, we have now cited this paper.

Reviewer 2 Report

In my knowledge, the term "wavefield" and the phrases like "wavefield process" are not applied for the electromagnetic or microwave scattering case (see lines 22, 26, 36, 42). The cases of US and MW scattering must be described in different terms. Line 112: abbreviation “PDA” did not define. Line 142: phrase “rough CSI reconstruction” is not standard one. It is better to use more understandable expressions. Sec. 2.2. It makes sense to discuss the need for the proximity condition of the reconstructed image CSI to the original image for the successful application of machine learning. Sec. 3: the title should be changed to “Numerical Experiments”. Lines 255 and 258: reference to Fig. 7 appears before that to Fig. 6 and both are before reference to Fig. 5. Fig. 4, Fig. 6 – Fig. 8: very small letters and numbers in these images. Both of Fig. 7 and Fig. 8 have unnecessary repetition of real and imaginary parts of the CSI reconstructions. Reference 1. Not correct reference: Vol. 208 is mistake for this book. References 19, 20. Missing year if issues.

Author Response

We wish to thank the reviewer for taking the time to evaluate our paper and for the thoughtful and constructive comments that have helped us to improve the paper. We have tried to address your comments as follows.

Once again, many thanks for your kind considerations.

In my knowledge, the term "wavefield" and the phrases like "wavefield process" are not applied for the electromagnetic or microwave scattering case (see lines 22, 26, 36, 42). The cases of US and MW scattering must be described in different terms.

Reply: We have now refrained from using the term “wavefield” in the manuscript as a shortcut to refer to either ultrasound or microwave imaging. The manuscript has been revised to refer to the specific modality we are referring to, either US, MW, or US/MW multimodal.

Abbreviation “PDA” did not define.

Reply: We thought that the acronym “PDE” was well-known as a short-cut for “Partial Differential Equations”. But, we agree with the reviewer that we should have defined this. It has now been done in the revised manuscript.

Phrase “rough CSI reconstruction” is not standard one. It is better to use more understandable expressions.

Reply: We have now removed this word.

It makes sense to discuss the need for the proximity condition of the reconstructed image CSI to the original image for the successful application of machine learning.

Reply:  We believe that we have addressed this concern in the subsection entitled “Qualitative Evaluation of Robustness”. Therein, we refer to Figures 5 and 6 where the CNN is tested with CSI images that were created after terminating the CSI algorithm in different numbers of iteration. As can be seen in these Figures, when the CSI algorithm is terminated after 20, 150, 250, and 500 iterations, the input images vary quite widely. Note that the training was performed with images created after 250 iterations. So we believe that if the reviewer is concerned with the “proximity” of the input images to the training set, then this discussion on the qualitative robustness should suffice. We have added a few sentences to this section to make this more clear. Also, we are providing numbers that for the RMS error for the real part of the permittivity case shown in Figure 5. We don’t believe it would be acceptable to add these numbers to the paper because this does not represent an extensive study of the robustness. That is why we called this section a “Qualitative Evaluation ...”. On the other hand, if the reviewer still thinks that we should add these numbers as a represntative case then we will.

Iteration 20

Iteration 150

Iteration 250

Iteration 500

Case 1

CSI(RMS)

2.2270

2.2216

2.2216

2.2250

CNN(RMS)

1.0117

1.0055

1.0055

1.0089

 Sec. 3: the title should be changed to “Numerical Experiments”.

Reply: We agree with the reviewer's comment and so we changed the title to numerical experiments.

Lines 255 and 258: reference to Fig. 7 appears before that to Fig. 6 and both are before reference to Fig. 5.

Reply: The authors thank the reviewer for his valuable suggestions. The mentioned problem has been solved in the revised version.

 Fig. 4, Fig. 6 – Fig. 8: very small letters and numbers in these images.

Reply: The authors thank the reviewer for his careful review. We increased the size of letter and numbers in these plots.

Both of Fig. 7 and Fig. 8 have unnecessary repetition of real and imaginary parts of the CSI reconstructions.

Reply: You are right, we removed that column of plots in the revised version.

Reference 1. Not correct reference: Vol. 208 is mistake for this book.

Reply: The authors thank the reviewer for his careful review and the error has been corrected.

References 19, 20. Missing year if issues. 

Reply: The authors thank the reviewer for his careful review and the error has been corrected.
